



# A soil moisture monitoring network to characterize karstic recharge and evapotranspiration at five representative sites across the globe

Romane Berthelin[1], Michael Rinderer[2], Bartolomé Andreo[3], Andy Baker[4], Daniela Kilian[5], Gabriele Leonhardt[5], Annette Lotz[5], Kurt Lichtenwoehrer[5], Matías Mudarra[3], Ingrid Y. Padilla[6], Fernando Pantoja Agreda[6], Rafael Rosolem[7], Abel Vale[8], Andreas Hartmann[1,7]

[1]Chair of Hydrological Modeling and Water Resources, Freiburg University, Freiburg, 79098, Germany
[2]Chair of Hydrology, Freiburg University, Freiburg, 79098, Germany
[3]Department of Geology and Centre of Hydrogeology. University of Malaga, Málaga, 29071, Spain
[4]Connected Waters Initiative Research Centre, UNSW, Sydney, NSW 2052, Australia
[5]Nationalpark Berchtesgaden, Berchtesgaden, 83471, Germany
[6]Department of Civil Engineering and Surveying, University of Puerto Rico, Mayagüez, 00682, Puerto Rico
[7]Department of Civil Engineering, University of Bristol, Bristol, BS8 1TR, United Kingdom
[8]Ciudadanos del Karso, 267 Sierra Morena PMB 230, San Juan, Puerto Rico 009264

*Correspondence to*: Romane Berthelin (*romane.berthelin@hydmod.uni-freiburg.de*)

**Abstract**

Karst systems that are characterized by a high subsurface heterogeneity are posing a challenge to study their complex recharge processes. Experimental methods to study karst processes mostly focus on characterizing the entire aquifer. Despite their important role for recharge processes, the limited focus has been given on studies of the soil and epikarst and most available research has been performed at sites of similar latitudes. In our study, we describe a new monitoring concept that allows the improvement of soil and epikarst processes understanding by covering different karst systems with different land cover at different climate regions. First, we describe the site selection and the experimental setup. Then we describe the five individual sites and their soil profiles. We also present some preliminary data and highlight the potential of the data for future research aimed at answering the relevant research questions: (1) How do the soil and epikarst heterogeneities influence water flow and storage processes in the karst vadose zone? (2) What is the impact of the land cover type on karstic groundwater recharge and evapotranspiration? (3) What is the impact of climate on karstic groundwater recharge and evapotranspiration? In order to answer these questions, we monitor soil moisture, which controls the partitioning of rainfall into infiltration, soil water storage, evapotranspiration, and groundwater recharge processes. We installed a soil moisture-monitoring network at five different climate regions: in Puerto Rico (tropical), Spain (Mediterranean), the United Kingdom (humid oceanic), Germany (humid mountainous), and Australia (dry semi-arid). At each of the five sites, we defined two 20m x 20m plots to install soil moisture probes under different land use types (forest and grassland). At each plot, 15 soil moisture profiles were installed with probes at different depths from the top soil to the epikarst (over 400 soil moisture probes were installed). Our first results show that the monitoring network provides new insights into the soil moisture dynamics of the five study sites and that significant differences prevail among forest and grassland sites. Some profiles are characterized by sequential reactions of soil moisture, i.e., the uppermost probe reacts first and the lowest probe reacts last, while at other profiles, we find non-sequential reactions that we interpret to result from preferential flow processes. While the former favours storage in the soil providing water for evapotranspiration, the latter can be seen as an indicator for the initiation of fast and preferential recharge into the karst system. Covering the spatiotemporal variability of these processes through a large number of installed probes, our monitoring network will allow to develop a new conceptual understanding of evapotranspiration and groundwater recharge processes in karst regions across different climate regions and land use types, and provide the base for quantitative assessment with physically-based modelling approaches in the future.



## 1 Introduction

Around a quarter of the global population fully or partially depends on water from karst aquifer (Ford and Williams 2013). In times of climate change, land use changes and growing population, this water availability is more and more vulnerable, both in term of water quantity and quality (Wada et al. 2010; Vörösmarty et al. 2010). Karst aquifers form through the chemical

dissolution of carbonate rocks. This process called karstification leads to subsurface heterogeneity expressed by features like conduits, caves, sinkholes, dolines (Hartmann et al. 2014). These geomorphologic heterogeneities lead to variable pathways and velocities that are highly variable in space and time. For that reason, karst aquifers have always posed a challenge in the characterization of karst hydrological processes for quantification of their present and future water storages (Goldscheider and Drew 2007).

Various experimental methods have been used in order to characterize karst systems. The most popular among them is the application of methods to analyze signals of artificial and natural tracers at the karst springs (Hartmann et al. 2014). Artificial tracers have been used to investigate the actual flow paths and flow times through karst systems (Mudarra et al. 2014; Goldscheider et al. 2008), while natural tracers, for instance, water isotopes, were used to understand the transit times and dispersion of water entering at the entire recharge of karst systems (Maloszewski et al. 2002). Continuous monitoring of karst

spring hydrographs and the hydrochemical signal have shown to reveal important information about the subsurface structure and dynamics of karst aquifers (Mudarra and Andreo 2011). Hydraulic methods like pumping tests revealed the local heterogeneity of hydraulic conductivities of karst aquifers (Giese et al. 2018) and geophysical methods allowed for obtaining information of geometrical information of the aquifer (Chalikakis et al. 2011) and thus the degree of development of the epikarst.

Generally applied by hydrogeologists, the abovementioned methods mostly focussed on characterizing the aquifer. Water recharge, one of the most important fluxes for balancing water budgets in karst aquifers, and the processes governing its dynamics in the vadose zone (Hartmann and Baker 2017), has received limited attention (Berthelin and Hartmann 2020). The vadose zone in karst systems is composed of the soil, the uppermost parts of the weathered carbonate rock (epikarst, Williams 1983) and the unsaturated carbonate rock (Hartmann and Baker 2017). Because of dissolution processes, the epikarst is highly

altered and presents dissolution features like Karren fields and fractures. The interface between soil and rock is then irregular and the porosity of rock decreases with depth. These characteristics lead to different hydrodynamic processes. Water in the shallow subsurface is subject to evapotranspiration. If there is soil, it can play an important role in infiltration velocity and mixing processes (Charlier et al. 2012; Perrin et al. 2003). The heterogeneous interface between soil and epikarst and the difference of permeability between both can allow the redistribution of the infiltrated water along the rock (Fu et al. 2015).

The difference of porosity between the high permeable rock of the epikarst and the non-weathered rock below can lead to the formation of perched aquifers (Williams 1983) and therefore to lateral redistribution of water flow through enlarged fractures. The rock and soil properties (porosity, fractures, lithology, karstification, etc…) are not the only factors that control recharge processes. The topography influences the water flow distribution, in particular, the epikarst topography, which can have a stronger control than the surface topography itself (Fu et al. 2015).  The thickness of the shallow subsurface also influences

the water flow velocity: a thin epikarst presents a higher proportion of large fractures and so a larger fraction of fast-flow components (Zhang et al. 2013). Likewise, the antecedent moisture conditions influence the storage capacity of water and infiltration rate. Under dry conditions, a larger portion of water can be stored, but subsurface flows are less important than under wet antecedent conditions (Charlier et al. 2012; Fu et al. 2015; Trček 2007). Finally, vegetation and climate influence karst recharge processes.

Approaches that were used to study soil and epikarst processes in the karst mostly relied on the above-mentioned methods to study the entire karst aquifer. Zhang et al. 2013 used springs discharge observations and studied the time lag of the hydrograph response to rainfall. They showed that the time lags are much longer in the thick epikarst zone than in the thin one and this for short, intermediate and long-term responses. Using hydrochemical methods such as major ions analyses and physiochemical



parameters, Houillon et al. 2017 characterize the flow conditions in the vadose zone, where piston and dilution mechanisms were identified. Isotope analysis of water and carbon as natural tracers confirmed that the epikarst highly influence karst recharge by providing important amounts of water in a fast and concentrated form from the epikarst zone to the rest of the system (Trček 2007). Indeed, these natural tracers can be used to investigate flow, mixing processes and residence times in

aquifers. Using artificial tracer tests, Kogovsek and Petric 2014 showed that depending on the tracer injection method, the former hydrological conditions and the geologic heterogeneities, a rapid infiltration can happen (hundred meters of bedrock in hours) but most of the tracer is in fact stored within the vadose zone and progressively flushed out in a period of several years. Aley and Kirkland 2012 showed the importance of considering horizontal flow in the processes occurring within the vadose zone by using a compilation of different artificial tracer tests studies. Just recently, Champollion et al. 2018 find that significant

water storage occurs in the first 10m of the vadose zone applying geophysical methods at two carbonate rock test sites in France. Most of these studies focussed on the traditional methods of karst characterization and were applied at individual sites at similar latitudes. Consequently, the derived understanding is difficult to transfer to other sites, for instance to other climate regions and more realistic conceptual models of karst recharge processes and evapotranspiration across different climates and land use types are still missing (Mudarra et al., 2019).

Here we present a soil moisture monitoring network to characterize karst recharge and evapotranspiration processes in five different climate regions, namely the tropical region (Puerto Rico), the Mediterranean region (Spain), the humid oceanic region (United Kingdom), the humid mountainous region (Germany) and the dry semi-arid region (Australia). Each site covered two land use types (grassland and forest). The information derived from soil moisture observations at a high spatial and temporal resolution of more than 400 soil moisture probes in five different climate regions will enable us to quantify the influence of

soil and epikarst heterogeneities, as well as land cover types, on the spatiotemporal dynamics of karstic recharge and evapotranspiration in these different climate conditions.

## 2 The monitoring concept

Our experimental concept is designed to answer the following questions:

1. How do the soil and epikarst heterogeneities influence water flow and storage processes in the karst vadose zone?
        2. What is the impact of the land cover type on the karst system recharge and evapotranspiration?
        3. What is the impact of climate on the karst system recharge and evapotranspiration?

To address these questions, we want to use standardized observations of soil moisture which has been shown in previous

studies to be informative to infer recharge processes (Ries et al. 2015).

### 2.1 Selection of sites

The experimental setup is designed to account for karstic heterogeneity and to enable the detection of lateral flow processes in the upper epikarst. It covers soil moisture dynamics of two types of land cover (grassland and forest) in five different climatic

regions. The consistent setup of soil moisture monitoring at all sites allows for comparison of the soil moisture information. Five study sites are selected across the globe (Figure 1) in order to cover distinct climate regions: tropical, Mediterranean, humid oceanic, humid mountainous, and dry semi-arid (Figure 1, Table 1). Two plots are selected on each site in order to study the impact of vegetation on recharge: one under grassland cover, the second one under forest. On each plot, 15 soil moisture monitoring profiles are installed resulting in 90 soil moisture probes at each site and a total number of 450 soil moisture probes

at all five sites. Several criteria are considered to choose the exact location of the plots. The two plots are chosen at locations





that are representative for typical grassland and forest land covers with similar slope and exposure. In order to fulfill these criteria, a GIS analysis was conducted using geological maps, and Digital Elevation Models to derive slope and exposure of the different studied karst systems. The following five sites are selected (see figure 1 and section 2.2 for a more detailed description of the sites):

1. Puerto Rico: El Tallonal, the northern part of the island. Tropical climate.
2. Spain: Villanueva del Rosario karst system, located 30 km north from Malaga. Mediterranean climate.
3. United-Kingdom, at the Sheepdrove Organic Farm in South of England on the West Berkshire Downs. Oceanic climate.
10   4. Germany, in the Berchtesgaden National Park, in the Northern limestones Alps. Humid continental climate.
5. Australia, above the Wellington Caves, at approximately 7km south of the town of Wellington (Southeast of Australia). Semi-arid climate. Soil moisture deficit can be observed most of the year because of a relatively high evaporation rates and low precipitation amount.

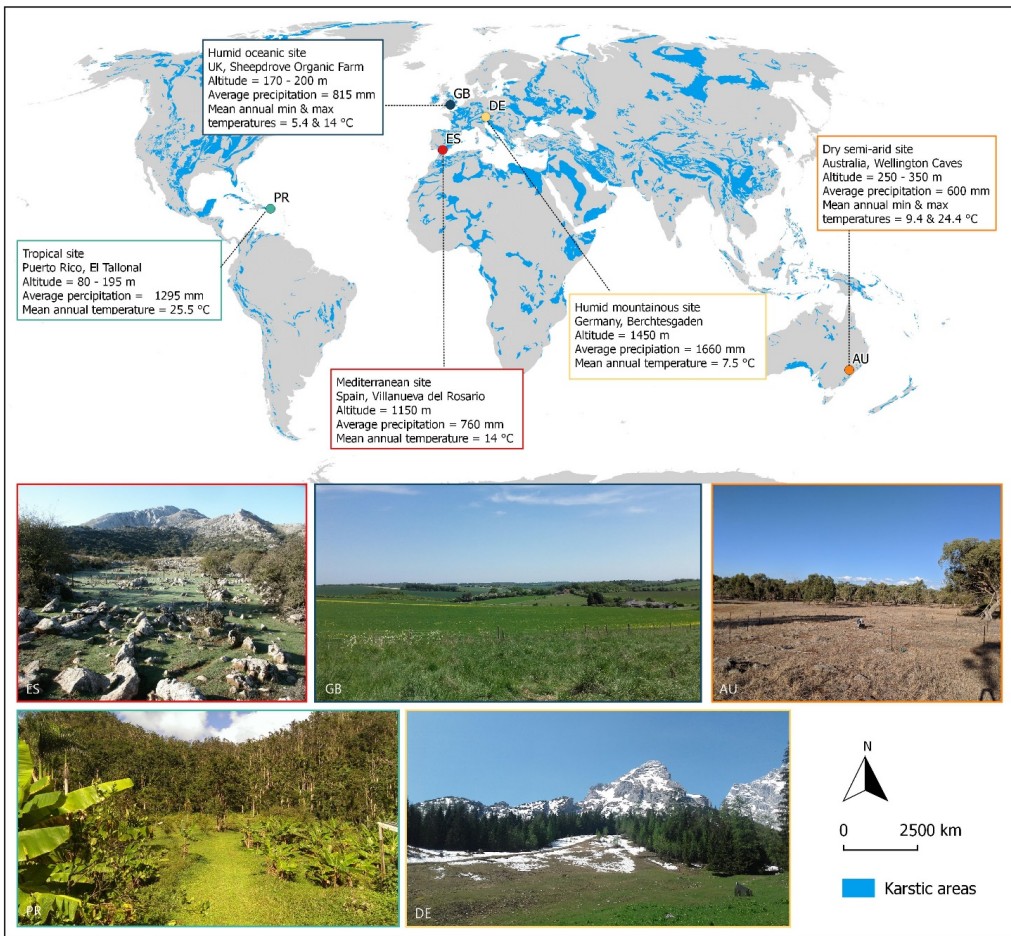

**Figure 1: Locations and pictures of the five sites - Carbonate rock outcrops derived from the World Karst Aquifer Map (Chen et al. 2017)**



## 2.2 Sites description

All of the selected sites are located at carbonate rock regions that have local or regional relevance for water resources management; they differ however in their surface and subsurface characteristics. A comparative overview is provided in Table 1 followed by detailed descriptions of the sites in the following subsections.

**Table 1: Overview of site properties - MAT = Mean Annual Temperature; MAP = Mean Annual Precipitation**

| | | Tropical region (PR) | Mediterranean region (ES) | Humide oceanic region (GB) | Humide mountainous region (DE) | Dry semi-arid region (AU) |
|---|---|---|---|---|---|---|
| Location | Zone | Private natural reserve - municipality of Arecibo- Puerto | Natural private property - 30 km north of Malaga - Spain | Land of a farm - West Berkshire Downs - England | National Park - Alps - Southeast Germany | Natural Reserve - 7,3 km south of Wellington - Australia |
| | Elevation | 80 - 195 m | 1150 m | 170 - 200 m | 1450 m | 250 - 350 m |
| | MAT | 25.5°C | 14 °C | 5.4 °C - 14 °C | 7.5°C | 9.4 °C – 24.4°C |
| | MAP | 1295 mm | 760 mm | 815 mm | 1660 mm | 617 mm |
| Climatic parameters (Koeppen-Geiger classification) | Climate | Tropical climate (Af) | Mediterranean climate (Csa) | Oceanic climate (Cfb) | Humid continental climate (Dfb) | Semi-arid climate (Bsh) |
| Vegetation | | Secondary forest with exotic plants as *Castilla elastic* and natives plants as *Guarea Guidonia* | Mediterranean scrubland with Mediterranean forest patches and pines from reforestation | Grass, hawthorn bushes, beech trees, cherry trees, maples | Grass, mountain pine and green alder shrubs | Native vegetation as Grassy White Box Woodland, exotic vegetation as Wild African Olive and broadleaf weeds |
| Geology | | Limestone from the late Oligocene period and early Miocene | Carbonate rocks from the Jurassic period (Limestones and dolostones) | White chalk from the upper Upper Cretaceous age | Triassic Dachstein limestone and Ramsau dolomite | Limestone of the middle Devonian |
| Hydrogeology | | The upper layer of the limestone aquifer: one of the most productive groundwater resources of the island | A fractured and karstified Jurassic carbonate aquifer - Villanueva del Rosario spring = 260 L/s | White chalk aquifer - groundwater table at tens of meters depth | Dynamic karst systems drained by around 330 springs | Marmorised, sub-vertical, fractured limestone containing hypogene caves, with some connectivity to adjacent alluvial aquifer |
| Soil | | Humid oxisol with minimal and simple development of horizons (Beinroth et al. 2003) | Patchy leptosols and silty-clayey texture soil (Marin et al. 2015) | Grey loamy soil with many flint stones and pieces of white chalk (Iwema 2017) | Rendzina (Nationalpark Berchtesgaden, 2001) | Red-brown soil comprising clays, iron oxides, fine quartz sands, and calcite nodules, with aeolian contribution (Rutlidge et al 2014) |

## 2.2.1 The tropical site (PR)

El Tallonal is a private natural reserve, located in the municipality of Arecibo, in the karstic zone, north of Puerto Rico

10  (18°40'62" N and 66°73'10"O). The average annual temperature is 25.5 °C; the annual precipitation is 1295 mm. According to the Holdridge life zones systems (Holdridge 1967), the reserve is classified as a subtropical moist forest zone (Ewel and Whitmore 1973). The dominant vegetation is characterized by secondary forest, where exotic plants as *Castilla elastica (S.)*, *Citrus spp. Musa sp (L.)* can be found, as well as native plants as *Guarea Guidonia (L.), Casearia sylvestris (Sw.), Casearia guianensis, Urtica dioica (L.), Roystonea borinquena (O.F.C)* (Fonseca da Silva 2014; Rivera-Sostre 2008). The geological

15  formation of the Tallonal reserve dates from the early Cretaceous period to the Quaternary era. The main karstic formation of the studied area is limestone of Aymamón and Aguada. These units date from the late Oligocene period and early Miocene. They are the result of several marine transgressions that happened during this period (Seiglie and Moussa 1984; Behrensmeyer





et al. 1992). Karst features are characterized by deep dolines, separated by high hills called mogotes, being the most rugged zone of the karstic stripe (Lugo et al. 2001). The watershed is hydrologically complex and is formed by the Rio Grande de Arecibo and the Tanamá river, bounded geologically by two subwatershed that discharge into the alluvial valley (Quiñones Aponte 1986). The upper layer of the northern limestone aquifer is located in the Aymamón and Aguada formations and contains one of the most productive groundwater resources of the island. Precipitation, surface streams, and runoff recharge the aquifer. In the mogotes zones, the recharge is mainly due to runoff during large precipitation events (Troester 1999). The soil of the Tallonal karstic area is defined as a humid oxisol with minimal and simple development of horizon according to the updated classification of Puerto Rico'soil (Beinroth et al. 2003). This type of soil is clayey, with a high water retention capacity, moderate fertility and high acidity that can limit the growth of some species of trees (Viera et al 2008).

Soil characterization in the field showed that roots in the soil of the grassland area concentrate in the first 15 cm of the soil in the grassland area where the texture is silty clay. Below 15 cm depth, the texture is more silt. We did not reach the bedrock at all grassland profiles. Thicknesses up to 9 m were reported by the landowner. At some profiles, we found deeper roots of *Andira inermis (W.)* and *Musa × paradisiaca (L.)* that are present all over the grassland plot. At the forest plot, we find an organic layer of more or less degraded organic matter of 20cm thickness. Below, the soil texture is clay and is getting more silt with the depth (from around 20 cm depth). A dense network of thick and thin roots is present on the entire profile; the maximum depth reached being around 30cm. The primary type of trees is *Coccoloba diversifolia (J.)*. Rock outcrops are present in the entire woodland plot.

### 2.2.2 The Mediterranean site (ES)

Villanueva del Rosario system (14 km$^2$ of catchment area) is a part of the Sierra Camarolos and Sierra del Jobo aquifer, located 30 km north of the city of Malaga in Spain, with altitudes from 600 to 1640 m (Marín et al. 2015). The mean historic annual precipitation is below 600 mm in the lower altitudes of the system, and more than 900 mm in the higher altitudes (Mudarra 2012). The average annual temperature is around 14 °C at 700 m above sea level (Mudarra 2012). The climate is Temperate Mediterranean. The vegetation is Mediterranean scrubland with Mediterranean forest patches and pines from reforestation (Marín et al. 2015). The geological formation of Villanueva del Rosario system is carbonate rocks from the Jurassic period, with a maximum thickness of 400 to 450 m (Peyre 1974). The rock is mainly composed of limestones and, in a lower proportion, dolostones mostly found below the limestone. The aquifer is fractured and karstified, bounded at almost all its tectonic borders by Upper Triassic clays and evaporate, Flysch clays and sandstones and Cretaceous-Paleogene marls (Mudarra et al. 2014). Karst features as karren fields, dolines, uvalas, and some sinkholes are present in the entire area. The system is drained by Villanueva del Rosario spring, located at the northern border of the system at 770 m altitude, which supplies drinking water to the village of Villanueva del Rosario. This spring has an annual mean flow discharge of 260 L/s and reacts rapidly to precipitation events (Mudarra et al. 2014).

The two main soil types developed at Villanueva del Rosario system are patchy leptosols that cover carbonates outcrops with a thickness lower than 30 cm, and silty-clayey texture soil with a thickness of 10 to 70 cm that covers the Cretaceous-Paleogene marls (Marín et al. 2015).

Soil characterization in the field showed that the soil in the grassland area can be divided into three horizons. The first one from ~ 0 to 15 cm depth has a silty texture. Most of the roots are concentrated at the first ~10 – 15 cm. The second horizon from ~ 15 to 40 cm has a silty clay texture. The third horizon from ~ 40 to 70 cm (maximum reached depth) has a clay texture. At some profiles, we found limestone rocks with a diameter of 10 to 30 cm. Shrubs and rock outcrops are present all over the grassland plot. At the forest plot, we can also divide the soil into 3 horizons. The first one from ~ 0 to 10 cm depth has a silty texture and organic content. The second horizon from ~ 10 to 35 cm has a silty texture. The third horizon from ~ 35 to 70 cm

(maximum reached depth) has a clay texture. Grass roots (in the top 10cm) and larger roots (with a diameter < 5cm) are abundant in most of the profiles. Trees (*Quercus ilex (L.)*, *Quercus faginea (L.),* and *Crataegus monogyna (J.)*) and rock outcrops are present in the entire woodland plot.

### 2.2.3 The humid oceanic site (GB)

The UK site is located on the property of the Sheepdrove Organic Farm, at the West Berkshire Downs (South England). The mean annual minimum and maximum temperature values are 5.4 °C and 14.0°C, respectively. The annual average precipitation is 815 mm. The climate is classified as Oceanic. The area is composed of meadow and forest with species such as hawthorn bushes, beech trees, cherry trees, maples (Iwema 2017). The study area is located in the Lambourn catchment, where the main geological formation is the Chalk of Upper Cretaceous age (Wheater et al. 2007). The white Chalk formation is the principal

aquifer of the region. It is highly permeable and karstified with dry valleys that run perpendicular to the Lambourn River, and has a groundwater table at tens of meters depth (Wheater et al. 2007; Rahman and Rosolem 2017).

The soil is a grey loamy soil with many Flintstones of up to 20 by 20 by 5 cm and pieces of white chalk (Iwema, 2017)

Soil characterization in the field showed that the soil at the grassland area can be divided into two horizons. The first ranges

from ~ 0 to 10 cm depth and is light brown with a silty texture with most of the roots concentrated in this layer. The second horizon starts from ~ 10 cm to various depths and is also light brown with a silty texture with pieces of chalk (diameter < 2 cm) and flint stones (diameter > 6cm). The occurrence of roots and flint stones decreases with depth and the number of chalk stones increases, such as the soil density, with depth.  White chalk bedrock appears at almost all profiles between ~ 30 cm to up to 60 cm depending on the location with a continuous transition between the soil and chalk is continuous. At the forest plot,

there is a 5 – 10 cm organic layer. Small roots are observed until ~10 cm depth. Thicker roots are also abundant across the entire soil depth at most of the profiles. A very soft chalk layer is found at almost all profiles at variable depths from ~ 10 cm to 35 cm.  The horizon from ~ 5 cm to the chalk layer has a silty texture. Some chalk stones (diameter < 2 cm) and flint stones (diameter > 6cm) can be found. The occurrence of chalk stones increases with the depth and leads to a continuous transition between the soil and chalk layer.

### 2.2.4 The humid mountainous site (DE)

The Berchtesgaden Land site is part of a National Park located in the Northern Limestone Alps in Southeast Germany. The average annual temperature is 7.5 °C and the annual average precipitation varies between 1500 and 2600 mm depending on the altitude. The minimum altitude in the national park is 603m with a maximum altitude at 2713. The climate is classified as humid continental.

The type of vegetation found in the park is grass, mountain pine and green alder shrubs (Garvelmann et al. 2017).

The dominant geological formation of the Berchtesgaden Land is Triassic Dachstein limestone and Ramsau dolomite. Jurassic and Cretaceous rock series are also present in the area. Karstification took place since the Alpine thrust exposed the limestone and typical karst features can be found as sinkholes, dry stream, caves, etc. Three valleys are drained by rivers from south to north that contribute to the Danube watershed (Kraller et al. 2011).  The karst systems that can be found on the area are drained

by around 330 springs, with a discharge from a few litres per second to several hundred litres per second. Most of them are located at the interface of the limestone and dolomite rock (Kraller et al. 2011). In the area of the National Park, three main types of soil can be found: Syrosem (35.5%), Cambisol (30.1%), and Podsol (26.7%) (Garvelmann et al. 2017).

The plots are located at an altitude of 1450 m with a mean annual precipitation of 1660 mm. The snow cover in this area is

present from mid of November to end of April/ beginning of May. Soil characterization in the field showed that the soil in the grassland area can be divided into three horizons. The first one from ~ 0 to 15 cm depth presents a high organic content.  Most



of the roots are concentrated at the first ~10 – 15 cm. At some profiles, roots can be found on the entire profile down to ~50 cm soil depth. The second horizon from ~ 15 to 50 cm has a silty clay texture. The third horizon from ~50 to 60 contains less clay than the previous depth. The second and third horizon thickness is highly variable depending on the profile location. Indeed, at the bottom of the soil profiles, from ~15 cm to ~65 cm, we find the bedrock composed of limestone rock pieces

(diameter <5 cm to >15 cm) mixed with sand. Some young growing pines (< 1m) are present all over the grassland plot. At the forest plot, we can also divide the soil into three horizons. The first one from ~ 0 to 20 cm depth has a high organic content. Grass roots (in the top 10cm) and larger roots (diameter < 5cm) are abundant in most of the profiles. The second horizon from ~ 20 to 30 cm has a clay texture. The third horizon from ~ 30 to 50 cm is getting brighter with a clay texture. The second and third horizon thickness is also highly variable depending on the profile location. The same rocky layer is observed at the bottom

of the soil profiles. Trees as Spruce and Larch and rocks are present in the entire woodland plot.

### 2.2.5 The dry semi-arid site (AU)

Wellington Caves site is a reserve located on the eastern side of the Catombal Range, and adjacent to the Bell River valley and alluvial aquifer, at 7.3 km south of the town of Wellington (148.936°E, 32.624°S). The Wellington Caves are an important

regional tourist resource. The annual average rainfall at Wellington is ~620 mm and the annual PET is ~1800 mm. The annual temperature ranges from around 0 to 45 °C with a maximum mean annual temperature of 24.3°C. The climate is classified as temperate semi-arid (Markowska et al. 2016).

The native Australian Grassy White Box Woodland dominates the vegetation, including the forest site. At the limestone outcrops, tree species such as *Brachychiton populneus (S. & E.)* (Kurrajong), *Callitris endlicheri (P.)* (Black Cypress), *Callitris*

*glaucophylla (T. & J.)* (White Cypress), *Eucalyptus albens (B.)* (White Box), can be found (Jex et al. 2012). Exotic vegetation is also present: Wild African Olive can be found at the immediate surface above the caves and has historically been present at the forest site.

The geology of the reserve is composed of deep-water marine sediments of the Ordovician Oakdale Formation and of limestone of the middle Devonian Garra Formation. The Oakdale Formation is present in a narrow band in the eastern part of the reserve.

The Garra Formation is present is the rest of the reserve. The Garra Formation limestone is locally marmorised, sub-vertically folded and composed of both massive and thinly bedded limestone with abundant marine fossils (Keshavarzi et al. 2017). Most of the significant karst features are developed in the massive limestone and have evidence of hypogene formation (Osborne 2007). Both the forest and grass sites are situated on the massive limestone.

The Bell River is located around 700 m west of the caves, with the river alluvium adjacent to the Garra limestone. The potential

connectivity between the river and the karst aquifer is high (Keshavarzi et al. 2017). Otherwise, no permanent streams can be found on the reserve. Adjacent to the forest site, and within Cathedral Cave, a network of drip loggers have been recording diffuse recharge at two depths (~4m and ~25 m) in the Garra Formation limestone since 2011. The logger network is described in Jex et al (2012), and time-series for diffuse recharge can be found in Cuthbert et al. (2014) and Markowska et al (2016). According to Jex et al. 2012, recharge happens if rainfall is at least around 60 mm within a 24-48h period, depending on soil

moisture antecedent conditions.

The soil in the Reserve is generally thin (<1m depth). Soil characterization in the field showed that at both, the forest and the grassland, the soils were backed-dry.

The soil in the grassland area presents most roots at the first ~5 cm. At deeper layers (below 5 cm), we find a sandy clay texture followed directly by carbonate rock in most of the cases. At some profiles, we found some larger rocks that could be part of

the soil skeleton or the beginning of the bedrock. At the woodland plot, we find a cover of partly degraded organic matter at the first ~5 cm. Below, until ~20 cm of depth, we find a clay soil and thicker roots. Below ~20 cm, an increasing fraction of





rocks appears indicating the beginning of the carbonate rock. Thick roots are abundant in most of the profiles. Trees (*Eucalyptus albens (B.)*) and rock outcrops are present in the entire woodland plot.

**2.3 Selection and setup of plots with contrasting land cover types**

At each site, we defined two plots of 400 m² each, in order to representatively capture the variability in epikarst heterogeneities.
A larger area would not allow capturing the spatial correlation of soil moisture dynamics between the soil moisture profiles; a smaller area would miss the spatial heterogeneity of epikarst that drives lateral soil water redistribution. At all plots, we defined squares of 20 m by 20 m, except for the UK grassland plot that is 32 meters by 12 meters due to restrictions of the landowner. 15 locations within each plot were randomly chosen for the installation of the soil moisture profiles (random sampling of profile locations from a uniform distribution) in order to cover a wide range of distances (i.e., potential characteristic length
scales) between the 15 profiles. At each profile, three probes at three different depths were installed, namely at 5 cm, 10 cm, and at the boundary between the soil and bedrock. The upper two probes are expected to measure the soil moisture signal that is predominantly controlled by vertical infiltration and evapotranspiration processes, while the soil moisture dynamics observed by the lowest probe can also be affected by lateral subsurface redistribution processes within the epikarst. The installation procedure did not allow installation at depths >80cm. In such cases, the probe was installed at this maximum depth
without reaching the epikarst.

Figure 2 shows the location of the grassland and forest plot for the Mediterranean climate site, with a schema that represents the distribution of the profile and their connection with the data logger. The two plots are close to each other in order to have similar weather conditions, slope, and exposure. This is the same for all sites, except for the UK, where the plots are ~1.5 km apart from each other due to restrictions of the landowner.

With the locations of all profiles defined, soil pits with a diameter of around 25 cm (to push the probes into the soil) and depth preferably down to the epikarst were drilled using hand tools and a gasoline driller which allowed maximum depths of ~80cm. For some few locations, we did not reach the epikarst (18; 23; 2 cases at the tropical climate site, the Mediterranean and the humid oceanic climate site, respectively). For some locations, the soil was too thin and fewer probes had to be installed (1; 1; 14 cases at the Tropical climate site, the humid oceanic and the dry semi-arid climate sites, respectively). During installation,
the characteristics of the soil and the orientation of the probes were recorded for each profile. The cables of the probes were protected by cable conduits and connected to the loggers, which are located at a central position of the plots (Figure 2). At each plot, we use two loggers in order to achieve manageable life times of the batteries and to avoid storage overflow of the loggers (see next subsection). In some cases, we applied additional protection measures at the plots. For example, a fence was necessary for the dry semi-arid climate grassland plot to prevent kangaroo damage. In the first months of logging, further
damages occurred from wild boars (Mediterranean climate site), martens (humid mountainous site) or humidity (tropical climate site), which required further protection measures.



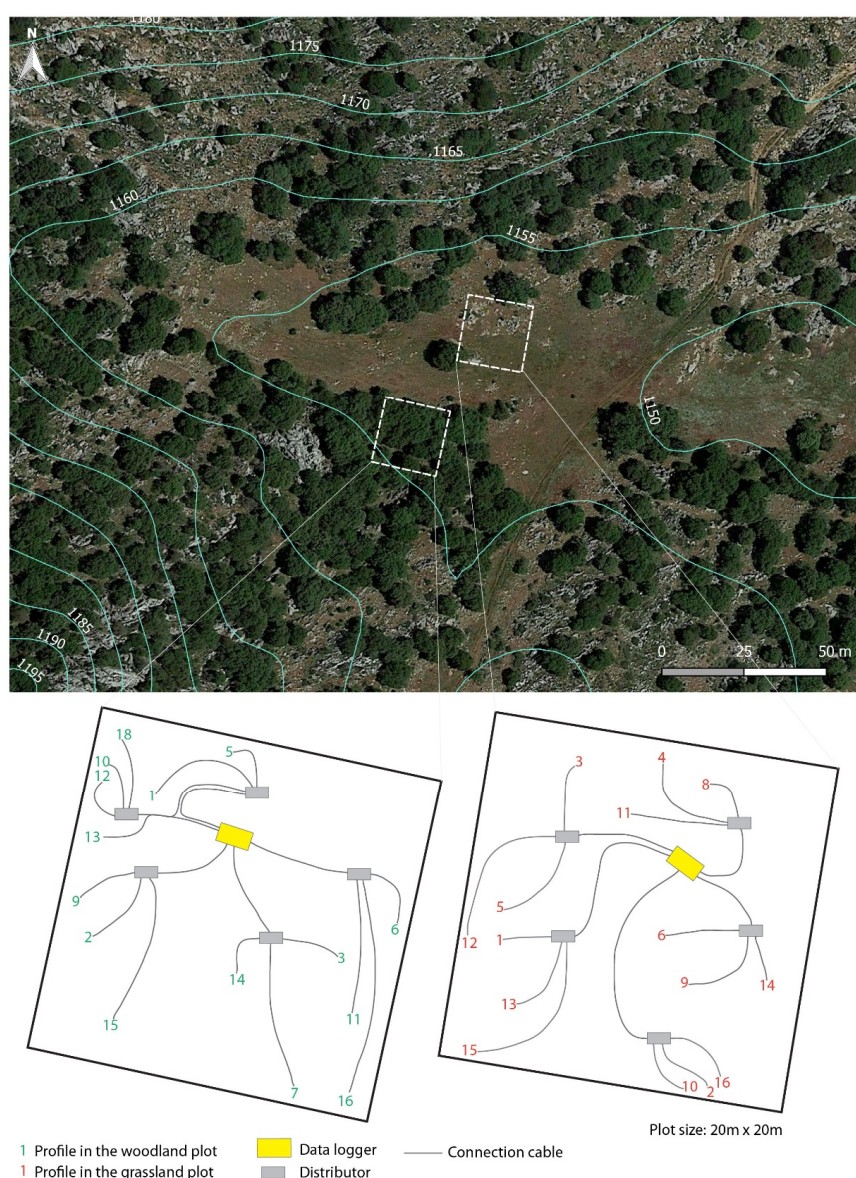

Figure 2: Plot locations, profile distributions, and cable connection maps at the Spanish site. The yellow boxes represent the data loggers. The grey boxes represent the distributors from the Truebner Company that allow the connection of 11 probes to one connection of the logger (one logger has four connections).

## 2.4 Technical description of probes and loggers

We used the soil moisture sensor "SMT100" from the company TRUEBNER (Figure 3), which combines the advantages of the Frequency-Domain-Reflectometry (FDR) sensor systems with the accuracy of a Time-Domain-Reflectometry (TDR) system. Its measurement is based on the travel time of a signal to determine the dielectric constant of the soil. For each of our plots, we used two "TrueLog100" (Figure 3) data logger, except for Puerto Rico where we used four loggers to enhance the battery life span. The data logger is powered with four D cell alkaline batteries and can be used for up to 50 sensors (in our case maximum 27). We programmed the data logger to take measurements with a time resolution of 15 minutes. This allows the coverage of rapid infiltration with a manageable life span of the batteries even during cold conditions (5-7 months) without





data storage overflow (320,000 measurements capacity, including water content, permittivity, counts, temperature and voltage supply in one measurement).

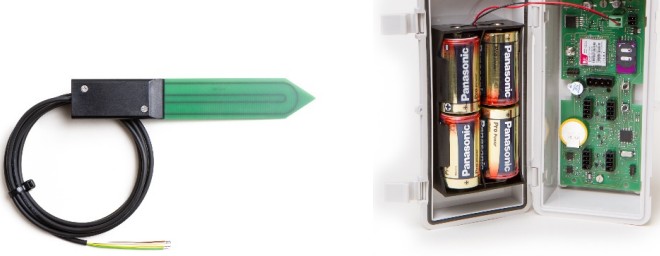

**Figure 3: Picture of the soil moisture sensors SMT100 and the data logger TrueLog100 from Truebner company**

The number of probes per site (~90) and in total (over 400) allows for new insights into the spatiotemporal variability of soil moisture dynamics at our sites. We, used the manufacturer's calibration and not a site specific calibration because we were mainly interested in the dynamics of soil moisture rather than their absolute values (see also Sprenger et al. 2015; Demand et al. 2019). Even without calibration, an observed increase of soil moisture will go along with infiltration as well as a plateau of very high soil moisture values can be seen as an indicator for saturation.

**3   First observations**

Figure 4 shows the first observations of soil moisture responses to individual rainfall events at selected profiles at the five sites; one profile at the grassland plot and one profile at the forest plot for each site. These examples show a broad range of soil moisture reactions. At the tropical site in the forest, we can observe that the deepest probe (21 cm soil depth) reacts strongest, in comparison to the two shallower ones (5 and 10 cm soil depth). We can also observe that a plateau is reached

which indicates soil saturation. Despite the differences of the soil properties at the grassland plot (subsection 2.2), we can observe a quite similar behavior as in the forest (with probes at depths 5 cm, 10 cm, and 50 cm). At the Mediterranean site at the grassland, the shallowest probe at 5 cm soil depth reacts first, then the second at 10 cm depth with some attenuation and the deepest probe (38 cm soil depth) does not react at all. However, at the forest plot, the deepest probe (34 cm soil depth) reacts stronger than the others do. Despite the dry period during which this first example was recorded, we still can observe a

response of soil moisture at the soil bedrock interface.

The profile shown for the humid oceanic grassland plot reacts very similar to the Mediterranean grassland plot: the first probe is the shallowest (5 cm soil depth), followed by the second at 10 cm soil depth. The deepest probe at the limit between the soil and the chalk (45 cm soil depth) is not reacting. The dynamics of the second probe are attenuated compared to the first. At the forest plot, the deepest probe (26 cm soil depth) is reacting stronger than the two shallower probes. At the humid mountainous

site, we find a consecutive reaction pattern at both the grassland plot and the forest plot with a first soil moisture reaction at the shallowest probe at 5 cm soil depth followed iteratively by the deeper probes (43 cm and 20 cm soil depth, respectively). At the grassland plot of the dry semi-arid site, the shallowest probe at 5 cm soil depth reacts first, followed by the second at 10 cm soil depth and the deepest one at 20 cm depth. An attenuation of the reaction is clearly observed with increasing depth. At the forest plot, the deepest probe in 15 cm depth reacts stronger than the shallowest ones in 5 and 10 cm soil depth.


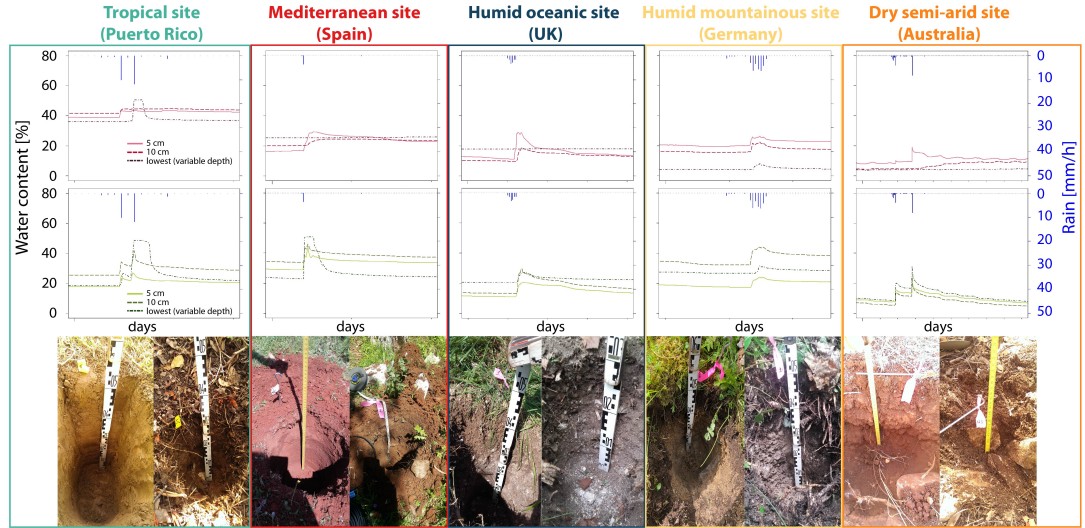

**Figure 4: Examples of observed soil moisture dynamics at each site (example events for selected profiles). The top graphs with the red shaded curves show the reaction of the probes at the grassland plot. The bottom graphs with the green shaded curves show the reaction of the probes at the forest plot. The photos on the bottom show the soil profiles of the respective plots (left: grassland, right**
**forested plot).**

Although still work in progress, this first observation of soil moisture dynamics, and their differences, already give a glance on the prevailing conditions of evapotranspiration and groundwater recharge at our different sites and plots. At some of the profiles, the deepest probes do not react. In such case, we can assume that a part of the water was evaporated or taken up by the vegetation, or the infiltrating water did not reach the lowest probe because of slow infiltration, water storage or lateral

redistribution. This could be the case of the reaction presented in figure 4 for the Mediterranean, humid oceanic and dry semi-arid grassland profile. At other profiles, we find a sequential reaction of all probes: the deepest probes react delayed compared to the upper ones and in an attenuated way. In this case, we could assume that infiltration is rather homogeneous (Demand et al. 2019) and that a part of the water is reaching the deeper subsurface to eventually contribute to groundwater recharge. This could be the case for the humid mountainous profiles presented in figure 4. On the other hand, we find non-sequential reactions

at other profiles, indicating a rather heterogeneous soil structure (Demand et al. 2019) with preferential flow path that can lead to rapid recharge dynamics (Ries et al. 2015) and quick discharge reactions (Chifflard et al. 2019) during heavy rainfall events.

## 4 Synthesis

In this work, we presented the objectives, measuring concept, and implementation of a soil moisture monitoring network at five representative sites across the globe to characterize karstic groundwater recharge and evapotranspiration. The monitoring

network aims at improving the understanding (1) of the influence of soil and epikarst heterogeneities in karst regions on the flow and storage processes in the karst vadose zone, (2) of climatic impacts on karstic groundwater recharge and evapotranspiration, and (3) of the impact of the land cover type on karstic groundwater recharge and evapotranspiration. In order to address these research objectives, >400 soil moisture probes were installed at five study sites located within different climatic regions. Each soil profiles consisted of 3 probes in 5 cm 10 cm and 80 cm depth (or at the soil bedrock interface, when

the soils were shallower). In order to account for the effect of different land covers, the profiles at each of the five sites were equally split between a grassland plot and a forest plot, each with an extent of 20m by 20m. Yet limited to a few rainfall-soil moisture events sampled so far, our data already reveal different soil moisture reactions at the different sites and land use types, such as sequential (top to bottom) reactions of soil moisture to rainfall events that favour evapotranspiration or non-sequential reactions indicating subsurface heterogeneity and preferential pathways that can result in enhanced groundwater recharge.



The wide range of local heterogeneity, land cover types and climate regions will allow new and detailed insights into the dynamics of groundwater recharge and evapotranspiration at the shallow subsurface in karstic regions. With longer records, additional data analysis will allow investigating the sequences of reaction of all available soil profiles (Demand et al. 2019), the identification of characteristic soil moisture states (Martini et al. 2015) and the quantitative assessment of soil hydraulic

properties and evapotranspiration and recharge rates by physically-based soil models (Sprenger et al. 2015) will be possible. The measuring network will provide comprehensive data to develop and test new conceptual models of the functioning of the soil and epikarst conditional to climate and land use (Enemark et al. 2019) that will help to improve the realism of water resources models for karst regions (Mudarra et al. 2019), the quantification of land-use change effects on karstic recharge (Sarrazin et al. 2018), or the simulation of above-cave hydrology for improved speleothem paleoclimate reconstruction

(Hartmann and Baker 2017, Cuthbert et al. 2014).

The monitoring network will collect data for at least three hydrological years. During that time, we plan to add more measurements to enhance our understanding of the karst shallow subsurface by collecting sample of stable soil water isotopes, water hydrochemistry groundwater dynamics (water level and karst spring discharges). At the end of the monitoring period, we will make our data publically available through a data publication in order to contribute to international efforts the Critical

Zone Observatories (Anderson et al. 2016), and to the International Soil Moisture Network (Dorigo et al. 2011) or initiatives to improve global earth systems models (Fan et al. 2019).

## 5 Author contributions

The research and monitoring program was designed and implemented by RB with advice from AH, MM and MR. The paper was conceived by RB guided by AH and MR. BA, AB, DK, GL, AL, KL, MM, IY.P, FPA, RR and AV provided their local

expertise before after and during the installation of the soil moisture network, as well as during the conception of the description of the study sites.

## 6 Acknowledgments

Romane Berthelin and Andreas Hartmann were supported by the Emmy Noether – Programme of the German Research Foundation (DFG; grant number HA 8113/1–1). Matías Mudarra and Bartolomé Andreo were supported by Group RNM-308

of Junta de Andalucía and Project CGL2015-65858-R. Rafael Rosolem is supported by the "A MUlti-scale Soil moisture Evapotranspiration Dynamics study" (AMUSED) [grant number NE/M003086/1] and the "Brazilian Experimental datasets for MUlti-Scale interactions in the critical zone under Extreme Drought" (BEMUSED) [grant number NE/R004897/1], both projects funded by Natural Environment Research Council (NERC). The article processing charge was funded by the German Research Foundation (DFG) and the University of Freiburg in the funding program Open Access Publishing. We thank Martin

Maier for his advice concerning the soil description.

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
