# Peer review of "A soil moisture monitoring network to characterize karstic recharge and evapotranspiration at five representative sites across the globe"

_Geoscientific Instrumentation, Methods and Data Systems, 2019_

## Referee Comment (RC1) · Anonymous Referee #1 · 28 Sep 2019

The paper introduces a major international field campaign and observational research program which is based on the networking of instruments for enhancing the high temporal and spatial resolution of soil moisture observations in karstic areas in different climates around the globe. This network will present novel data that could help to develop physical and conceptual models in the karstic regions. The strategy for selecting different climates, land-uses, scientific methods, and assumptions are valid and clearly outlined. The description of experiments and calculations is sufficiently complete and precise to allow their reproduction by fellow scientists. The authors give proper credit

to related work and clearly indicate their own new/original contribution. The language is fluent and precise; and symbols, abbreviations, and units are correctly defined and used. The amount and quality of supplementary material and references are appropriate. The title clearly reflects the contents of the paper. The overall presentation is well structured and clear. The results are not sufficient to support the interpretations and conclusions, however, they are useful for giving positive indications about the correctness of authors' assumptions. The abstract provides a complete summary but is not concise and lacks the keywords.

---

## Referee Comment (RC2) · Anonymous Referee #2 · 21 Oct 2019

This paper propose an interesting network for monitoring soil humidity in contrasted karst area. This experiments have a high potential to understand subsurface flow in karst and I recommend the publication. With 5 places in the world, the ambition to discuss impact of climate on recharge seems unrealistic, but the device is well design to validate models with a dataset including a large range of variability/configurations. Introduction: At the system scale, rainfall/discharge models are the one of the most popular method to quantify the karst recharge. This should be mentioned with ex- amples. The aims of this paper is to propose a soil moisture monitoring network in

karst area. This is not the first time that the soil moisture is monitored in karst area, I know some examples in China, in Houillon et al. 2016 as part of the SNO KARST in France with interesting results. This should be mentioned in your introduction. L11 Hartmann et al. 2014 are not the first to use tracer in karst system, if it is the most popular approach, older references should exist. I wonder if all the selected sites are included in areas where precipitation and ET0 are known? Overwise Future models of soil moisture partitioning will have low constraint. To fully valorize a 15 min sampling rate, the rain gauge should be close to the monitoring device and have the same sampling rate. In the same way concerning valorization of this dataset, are this site located in catchment where regional karst spring is also monitored? This should open nice confrontations between entire aquifer approaches and the proposed one. This could be mentioned in the manuscript. Slope is one of the main driver to describe infiltration in soil, as mentioned P4L2. The value of the slope are not given. Are all the site located in the same range of slope value? Photo suggest that site are all in flat areas? In the same way, the geomorphologic location can be a main driver: upper part of a plateau or depression areas where preferential recharge take place? This could be mentioned. Soil description is mainly qualitative, what about bulk density, porosity, conductivity. The observed (non)sequential reactions can be explain by properties heterogeneity with deep. At this scale, the decrease of properties with depth suggested in introduction is not obvious and should be validated site by site. If you choose to present monitoring network into a GI paper, without waiting for more results, this suggest that this network is design for a long time. In conclusion, you speak about 3 years, it is surprising.

---

## Short Comment (SC1) · 13 Nov 2019

The paper introduces a major international field campaign and observational research program which is based on the networking of instruments for enhancing the high temporal and spatial resolution of soil moisture observations in karstic areas in different climates around the globe. This network will present novel data that could help to develop physical and conceptual models in the karstic regions. The strategy for selecting different climates, land-uses, scientific methods, and assumptions are valid and clearly outlined. The description of experiments and calculations is sufficiently complete and

precise to allow their reproduction by fellow scientists. The authors give proper credit to related work and clearly indicate their own new/original contribution. The language is fluent and precise; and symbols, abbreviations, and units are correctly defined and used. The amount and quality of supplementary material and references are appropriate. The title clearly reflects the contents of the paper. The overall presentation is well structured and clear.

Reply: We thank the reviewer for her/his positive and constructive comments that will contribute to improve the manuscript. According to her/his comments, we will perform the following changes.

The results are not sufficient to support the interpretations and conclusions, however, they are useful for giving positive indications about the correctness of authors' assumptions.

Reply: We will re-write the conclusions in order to provide outlooks instead of interpretations, which are indeed not strong enough given the short examples that we can provide at this stage.

The abstract provides a complete summary but is not concise and lacks the keywords.

Reply: We will shorten the abstract and rephrase it in a more comprehensive way. The missing keywords will be added.

---

## Short Comment (SC2) · 13 Nov 2019

This paper propose an interesting network for monitoring soil humidity in contrasted karst area. This experiments have a high potential to understand subsurface flow in karst and I recommend the publication.

Reply: We thank the reviewer for his/her useful and valuable comments that will help to improve the manuscript. We will apply all suggestions and clarify points as suggested in her/his comments.

With 5 places in the world, the ambition to discuss impact of climate on recharge seems unrealistic, but the device is well design to validate models with a dataset including a large range of variability/configurations.

Reply: Indeed, the quantitative comparison of the impact of climate on recharge will be questionable as too many different parameters can have an influence apart the climate. Instead of focussing on differences among the sites caused by climate, the network allows to analyse differences that are caused by the integrated influence of different parameters. However, the repetition of the measurements settings at each site will at least allow a qualitative comparison. We will clarify the respective formulations in the manuscript accordingly.

Introduction: At the system scale, rainfall/discharge models are the one of the most popular method to quantify the karst recharge. This should be mentioned with examples.

Reply: We will mention rainfall/discharge models used to quantify the karst recharge including and provide the respective references.

The aims of this paper is to propose a soil moisture monitoring network in karst area. This is not the first time that the soil moisture is monitored in karst area, I know some examples in China, in Houillon et al. 2016 as part of the SNO KARST in France with interesting results. This should be mentioned in your introduction.

Reply: We will add further the literature and consider the suggested examples to mention previous work using soil moisture in karst systems.

L11 Hartmann et al. 2014 are not the first to use tracer in karst system, if it is the most popular approach, older references should exist.

Reply: We apologize for this confusion, Hartmann et al. 2014 is a review paper including diverse references using tracer in karst system. We will precise it in the text and add other references.

I wonder if all the selected sites are included in areas where precipitation and ET0 are known? Overwise Future models of soil moisture partitioning will have low constraint. To fully valorize a 15 min sampling rate, the rain gauge should be close to the monitoring device and have the same sampling rate.

Reply: We thank the reviewer to point this out. Indeed, climate stations are installed close to all our sites. Since the stations are operated by our collaboration partners, their temporal resolution differs due to energy supply or storage limitations. For the site in Puerto Rico the measurement rate is 5 min. For the sites in Germany and Australia, the rate measurement is 10 min. For the one in Spain it is 30 min and 1 hour for the one in the UK. We will provide this information and discuss possible consequences of varying temporal resolution in the revised version of the manuscript.

In the same way concerning valorization of this dataset, are this site located in catchment where regional karst spring is also monitored? This should open nice confrontations between entire aquifer approaches and the proposed one. This could be mentioned in the manuscript.

Reply: This valuable comment was already addressed to us in the past. For that reason, CTD diver probes measuring water levels were recently installed at the sites drained by a spring. No data has yet been downloaded but we will mention the value of observing the aquifer response in addition to the soil moisture monitoring plots in the manuscript as suggested.

Slope is one of the main driver to describe infiltration in soil, as mentioned P4L2. The value of the slope are not given. Are all the site located in the same range of slope value? Photo suggest that site are all in flat areas? In the same way, the geomorphologic location can be a main driver: upper part of a plateau or depression areas where preferential recharge take place? This could be mentioned.

Reply: Almost all sites are at locations with similar slope. Except for the German site, as it is located in a mountainous area with steeper slopes, which are typical for mountain karst regions. We will add the slope values in the description table. Concerning the geomorphological location, most of the sites are located at slopes close to the plateau of the karst system in order to avoid disturbance by groundwater discharge. The respective information will be provided in the revised manuscript.

Soil description is mainly qualitative, what about bulk density, porosity, conductivity.

Reply: Soil texture analyses are currently conducted in the laboratory and we will add their results to the revised version of the manuscript. Due the difficulty to transport the soil samples from the sites back to Freiburg, bulk density, porosity, or conductivity measurements were not possible.

The observed (non)sequential reactions can be explain by properties heterogeneity with deep. At this scale, the decrease of properties with depth suggested in introduction is not obvious and should be validated site by site.

Reply: We will clarify this point in the manuscript. Preferential flow (non-sequential reactions) might take place at different places or not. It is a hypothesis based on literature that we will use for future analysis. As our currently available data is not complete enough for detailed analysis, we will replace our interpretations on non-sequential reactions with a short outlook (also see our 2nd reply to reviewer #1).

If you choose to present monitoring network into a GI paper, without waiting for more results, this suggest that this network is design for a long time. In conclusion, you speak about 3 years, it is surprising.

Reply: The 3 years represent the temporal frame of the PhD project that will be the first to work on the collected data. We hope that the monitoring network will continue to be maintained in the future depending on the collaborators' possibilities and interest. We will clarify this point in the manuscript to avoid confusion.

---

## Author Response (AR1)

CHAIR OF HYDROLOGICAL MODELING     Friedrichstrasse 39
AND WATER RESOURCES     D -79098 Freiburg

GI Editorial board

**Romane Berthelin**

PhD student

Chair of **Hyd**rological **Mod**eling and Water Resources
University of Freiburg

Telefon: +49 (0)761-203-69245
Telefax: +49 (0)761-203-3594
romane.berthelin@hydmod.uni-freiburg.de
www.hydmod.uni-freiburg.de

Friedrichstrasse 39
D-79098 Freiburg

Freiburg, December 16th 2019

Dear Salvatore Grimaldi,

Thank you for your email letter of November 16th 2019 to inform us about the closure of the open discussion of our manuscript "A soil moisture network to characterize karstic recharge and evapotranspiration at five representative sites across the globe" with the reference gi-2019-22.

Our detailed response to all comments can be found in the point-by-point response just below.

Consequently, we are pleased to provide a thoroughly revised manuscript. We would like to thank the two reviewers for their valuable set of recommendations. These have allowed us to improve the manuscript. We hope that the revised version is now satisfactory for publication in Geoscientific Instrumentation, Method and Data Systems.

Sincerely,

Romane  Berthelin
(on behalf of all the authors)

**Referee #1**

Reply to comments by an anonymous #1 on the manuscript „A soil moisture monitoring network to characterize karstic recharge and evapotranspiration at five representative sites across the globe" by Berthelin et al.

The paper introduces a major international field campaign and observational research program which is based on the networking of instruments for enhancing the high temporal and spatial resolution of soil moisture observations in karstic areas in different climates around the globe. This network will present novel data that could help to develop physical and conceptual models in the karstic regions. The strategy for selecting different climates, land-uses, scientific methods, and assumptions are valid and clearly outlined. The description of experiments and calculations is sufficiently complete and precise to allow their reproduction by fellow scientists. The authors give proper credit to related work and clearly indicate their own new/original contribution. The language is fluent and precise; and symbols, abbreviations, and units are correctly defined and used. The amount and quality of supplementary material and references are appropriate. The title clearly reflects the contents of the paper. The overall presentation is well structured and clear.

*Reply: We thank the reviewer for her/his positive and constructive comments that contributed to improve the manuscript. According to her/his comments, we performed the following changes.*

The results are not sufficient to support the interpretations and conclusions, however, they are useful for giving positive indications about the correctness of authors' assumptions.

*Reply: We specified in the discussion that the work is on progress and that the interpretations made from the reaction examples are only the first suggestions. They should be explored in more detail in future work. The focus of this paper is to present the experimental design. The data that we showed are used as examples to illustrate the future possibilities that the data will give in the exploration of the different flow processes happening during karst recharge.*

*See page 13 line 6 of the revised manuscript.*

The abstract provides a complete summary but is not concise and lacks the keywords.

*Reply: We shortened the abstract. The following keywords were added below the revised abstract: soil moisture, karst characterization, groundwater recharge, evapotranspiration.*

**Referee #2**

Reply to comments by an anonymous referee #2 on the manuscript „A soil moisture monitoring network to characterize karstic recharge and evapotranspiration at five representative sites across the globe" by Berthelin et al.

This paper propose an interesting network for monitoring soil humidity in contrasted karst area. This experiments have a high potential to understand subsurface flow in karst and I recommend the publication.

*Reply: We thank the reviewer for his/her useful and valuable comments that helped to improve the manuscript. We applied the following suggestions as suggested in her/his comments.*

With 5 places in the world, the ambition to discuss impact of climate on recharge seems unrealistic, but the device is well design to validate models with a dataset including a large range of variability/configurations.

*Reply: Indeed, the quantitative comparison of the impact of climate on recharge will be questionable as too many different parameters can have an influence apart from the climate. Instead of focussing on differences among the sites caused by climate, the network allows analyzing differences that are caused by the integrated influence of different parameters. However, the repetition of the measurements settings at each site will at least allow a qualitative comparison. We clarified the respective formulations in the manuscript accordingly (see page 3 line 24 and 32 and page 13 line 25 of the revised manuscript).*

Introduction: At the system scale, rainfall/discharge models are the one of the most popular method to quantify the karst recharge. This should be mentioned with examples.

*Reply: We mentioned rainfall/discharge models used to quantify the karst recharge and provided the respective references (see the introduction of the revised manuscript: page 2 line 14).*

The aims of this paper is to propose a soil moisture monitoring network in karst area. This is not the first time that the soil moisture is monitored in karst area, I know some examples in China, in Houillon et al. 2016 as part of the SNO KARST in France with interesting results. This should be mentioned in your introduction.

*Reply: Houillon et al. 2017 do not seem to use soil moisture in their study. However, we used this reference to discuss the use of hydrochemical methods and physicochemical parameters to characterize the flow conditions in the vadose zone. We also added the reference of a study conducted in China that investigates soil moisture in a karst area to compare the impact of vegetation on soil water storage. See page 2 line 40.*

L11 Hartmann et al. 2014 are not the first to use tracer in karst system, if it is the most popular approach, older references should exist.

*Reply: We apologize for this confusion, Hartmann et al. 2014 is a review paper including diverse references using tracer in karst system. We clarified this in the text. Other references of studies using tracers are cited in the following text (see page 2 line 6 of the revised manuscript).*

I wonder if all the selected sites are included in areas where precipitation and ET0 are known? Overwise Future models of soil moisture partitioning will have low constraint. To fully valorize a 15 min sampling rate, the rain gauge should be close to the monitoring device and have the same sampling rate.

*Reply: We thank the reviewer for pointing this out. Indeed, climate stations are installed close to all our sites. Since the stations are operated by our collaboration partners, their temporal resolution differs due to energy supply and storage limitations. We provided information about the precipitation measurements on page 10 line 30 of the revised manuscript.*

In the same way concerning valorization of this dataset, are this site located in catchment where regional karst spring is also monitored? This should open nice confrontations between entire aquifer approaches and the proposed one. This could be mentioned in the manuscript.

*Reply: This valuable suggestion was already given to us by other colleagues in the past. For that reason, CTD diver probes measuring water levels are currently being installed at the sites drained by a spring. No data has yet been downloaded. We added this information on page 10 line 33 and page 13 line 18 in the revised manuscript.*

Slope is one of the main driver to describe infiltration in soil, as mentioned P4L2. The value of the slope are not given. Are all the site located in the same range of slope value? Photo suggest that site are all in flat areas? In the same way, the geomorphologic location can be a main driver: upper part of a plateau or depression areas where preferential recharge take place? This could be mentioned.

*Reply: Almost all sites are at locations with a similar slope. Except for the German site, which is located in a mountainous area with steeper slopes, which are typical for mountain karst regions. We added the slope values in the description table 1.*

Soil description is mainly qualitative, what about bulk density, porosity, conductivity.

*Reply: Soil texture analyses were conducted in the laboratory for at least one soil sample per site. We added the results to the revised version of the manuscript. Due to the difficulty to transport the soil samples from the sites back to Freiburg, bulk density, porosity, or conductivity measurements were not possible.*

The observed (non)sequential reactions can be explain by properties heterogeneity with deep. At this scale, the decrease of properties with depth suggested in introduction is not obvious and should be validated site by site.

*Reply: Preferential flow (non-sequential reactions) might take place at different places or not. It is a hypothesis based on literature that we will use for future analysis. As answered to reviewer*

*#1, we specified in the discussion that the work is still in progress and that the interpretations made from the reaction examples are only the first suggestions to be explored in more detail in the future. See our response to the 2nd remark of review #1 and the respective changes from page 13 line 6.*

If you choose to present monitoring network into a GI paper, without waiting for more results, this suggest that this network is design for a long time. In conclusion, you speak about 3 years, it is surprising.

*Reply: The 3 years represent the temporal frame of the PhD project that will be the first to work on the collected data. We hope that the monitoring network will continue to be maintained in the future depending on the collaborators' possibilities and interests. We clarified this point in the manuscript to avoid confusion. See page 14 line 15 in the revised manuscript.*

**A soil moisture monitoring network to characterize karstic recharge and evapotranspiration at five representative sites across the globe**

Romane Berthelin[1], Michael Rinderer[2], Bartolomé Andreo[3], Andy Baker[4], Daniela Kilian[5], Gabriele Leonhardt[5], Annette Lotz[5], Kurt Lichtenwoehrer[5], Matías Mudarra[3], Ingrid Y. Padilla[6], Fernando Pantoja Agreda[6], Rafael Rosolem[7], Abel Vale[8], Andreas Hartmann[1,7]

[1]Chair of Hydrological Modeling and Water Resources, Freiburg University, Freiburg, 79098, Germany
[2]Chair of Hydrology, Freiburg University, Freiburg, 79098, Germany
[3]Department of Geology and Centre of Hydrogeology. University of Malaga, Málaga, 29071, Spain
[4]Connected Waters Initiative Research Centre, UNSW, Sydney, NSW 2052, Australia
[5]Nationalpark Berchtesgaden, Berchtesgaden, 83471, Germany
[6]Department of Civil Engineering and Surveying, University of Puerto Rico, Mayagüez, 00682, Puerto Rico
[7]Department of Civil Engineering, University of Bristol, Bristol, BS8 1TR, United Kingdom
[8]Ciudadanos del Karso, 267 Sierra Morena PMB 230, San Juan, Puerto Rico 009264

*Correspondence to*: *Romane Berthelin (romane.berthelin@hydmod.uni-freiburg.de)*

**Abstract**

Karst systems  are characterized by a high subsurface heterogeneity and their complex recharge processes are difficult to characterize . Experimental methods to study karst systems mostly focus on analysing the entire aquifer. Despite their important role for recharge processes,  the soil and epikarst received limited attention  and the few available studies were performed at sites of similar latitudes. In this paper, we describe a new monitoring network that allows the improvement of soil and epikarst processes understanding by covering different karst systems with different land cover at different climate regions. Here, we  present  preliminary data of the network and elaborate their potential to answer research questions about the role of soil and epikarst on karstic water flow and storage . The network measures soil moisture at multiple points and depths to understand the partitioning of rainfall into infiltration,  evapotranspiration, and groundwater recharge processes. We installed  soil moisture probes at five different climate regions: Puerto Rico (tropical), Spain (Mediterranean), the United Kingdom (humid oceanic), Germany (humid mountainous), and Australia (dry semi-arid). At each of the five sites, we defined two 20m x 20m plots with different land use types (forest and grassland). At each plot, 15 soil moisture profiles were randomly selected and  probes at different depths from the top soil to the epikarst (in total over 400 soil moisture probes) were installed. Covering the spatiotemporal variability of flow processes through a large number of profiles, our monitoring network will allow to develop a new conceptual understanding of evapotranspiration and groundwater recharge processes in karst regions across different climate regions and land use types, and provide the base for quantitative assessment with physically-based modelling approaches in the future.

Keywords: soil moisture, karst characterization, groundwater recharge, evapotranspiration

[revised manuscript text omitted]

At each site, a climate station is installed close to the plots. Precipitation is measured at different temporal resolutions depending on the site dependent on energy supply and storage capacities. For the tropical climate site, the temporal resolution is 5 min. For the humid mountainous and dry semi-arid sites, the

35  resolution is 10 min. For the Mediterranean one, it is 15 min and 1 h for the  humid oceanic climate site.

Discharge is monitoreing at some of the sites (tropical, Mediterranean, humid mountainous and dry semi-arid ). Yet not available, these data will allow the determination of the link between the discharge and the soil moisture signal.

[Figure]

**Figure 2: Plot locations, profile distributions, and cable connection maps at the Spanish site. The yellow boxes represent the data loggers. The grey boxes represent the distributors from the Truebner Company that allow the connection of 11 probes to one connection of the logger (one logger has four connections).**

5   **2.4 Technical description of probes and loggers**

We used the soil moisture sensor "SMT100" from the company TRUEBNER (Figure 3), which combines the advantages of the Frequency-Domain-Reflectometry (FDR) sensor systems with the accuracy of a Time-Domain-Reflectometry (TDR) system. Its measurement is based on the travel time of a signal to determine the dielectric constant of the soil. For each of our plots, we used two "TrueLog100" (Figure 3) data logger, except for Puerto Rico where we used four loggers to enhance the

10   battery life span. The data logger is powered with four D cell alkaline batteries and can be used for up to 50 sensors (in our case maximum 27). We programmed the data logger to take measurements with a time resolution of 15 minutes. This allows the coverage of rapid infiltration with a manageable life span of the batteries even during cold conditions (5-7 months) without

data storage overflow (320,000 measurements capacity, including water content, permittivity, counts, temperature and voltage supply in one measurement).

[Figure]

**Figure 3: Picture of the soil moisture sensors SMT100 and the data logger TrueLog100 from Truebner company**

5 The number of probes per site (~90) and in total (over 400) allows for new insights into the spatiotemporal variability of soil moisture dynamics at our sites. We, used the manufacturer's calibration and not a site specific calibration because we were mainly interested in the dynamics of soil moisture rather than their absolute values (see also Sprenger et al. 2015; Demand et al. 2019). Even without calibration, an observed increase of soil moisture will go along with infiltration as well as a plateau of very high soil moisture values can be seen as an indicator for saturation.

10 **3    First observations**

Figure 4 shows the first observations of soil moisture responses to individual rainfall events at selected profiles at the five sites; one profile at the grassland plot and one profile at the forest plot for each site. These examples show a broad range of soil moisture reactions. At the tropical site in the forest, we can observe that the deepest probe (21 cm soil depth) reacts strongest, in comparison to the two shallower ones (5 and 10 cm soil depth). We can also observe that a plateau is reached

15 which indicates soil saturation. Despite the differences of the soil properties at the grassland plot (subsection 2.2), we can observe a quite similar behavior as in the forest (with probes at depths 5 cm, 10 cm, and 50 cm). At the Mediterranean site at the grassland, the shallowest probe at 5 cm soil depth reacts first, then the second at 10 cm depth with some attenuation and the deepest probe (38 cm soil depth) does not react at all. However, at the forest plot, the deepest probe (34 cm soil depth) reactsresponds stronger than the others do. Despite the dry period during which this first example was recorded, we still can

20 observe a response of soil moisture at the soil bedrock interface.

The profile shown for the humid oceanic grassland plot reacts very similar to the Mediterranean grassland plot: the first probe is the shallowest (5 cm soil depth), followed by the second at 10 cm soil depth. The deepest probe at the limit between the soil and the chalk (45 cm soil depth) is not responding. The dynamics of the second probe are attenuated compared to the first. At the forest plot, the deepest probe (26 cm soil depth) is reacting stronger than the two shallower probes. At the humid

25 mountainous site, we find a consecutive reaction pattern at both the grassland plot and the forest plot with a first soil moisture response at the shallowest probe at 5 cm soil depth followed iteratively by the deeper probes (43 cm and 20 cm soil depth, respectively). At the grassland plot of the dry semi-arid site, the shallowest probe at 5 cm soil depth reacts first, followed by the second at 10 cm soil depth and the deepest one at 20 cm depth. An attenuation of the reaction is clearly observed with increasing depth. At the forest plot, the deepest probe in 15 cm depth responds stronger than the shallowest ones in 5 and

30 10 cm soil depth.

[Figure]

**Figure 4: Examples of observed soil moisture dynamics at each site (example events for selected profiles). The top graphs with the red shaded curves show the reaction of the probes at the grassland plot. The bottom graphs with the green shaded curves show the reaction of the probes at the forest plot. The photos on the bottom show the soil profiles of the respective plots (left: grassland, right forested plot).**

Although still work in progress, the observations shown above already give a glance on the different flow processes that might happen at our sites and plots.  At some of the profiles, the deepest probes do not react. In such case, we can assume that a part of the water was evaporated or taken up by the vegetation, or the infiltrating water did not reach the lowest probe because of slow infiltration, water storage or lateral redistribution. This could be the case of the response presented in figure 4 for the Mediterranean, humid oceanic and dry semi-arid grassland profile. At other profiles, we find a sequential reaction of all probes: the deepest probes respond delayed compared to the upper ones and in an attenuated way. In this case, we could assume that infiltration is rather homogeneous (Demand et al. 2019) and that a part of the water is reaching the deeper subsurface to eventually contribute to groundwater recharge. This could be the case for the humid mountainous profiles presented in figure 4. On the other hand, we find non-sequential reactions at other profiles, indicating a rather heterogeneous soil structure (Demand et al. 2019) with preferential flow path that can lead to rapid recharge dynamics (Ries et al. 2015) and quick discharge responses (Chifflard et al. 2019) during heavy rainfall events. Still at the stage of preliminary analyses, these potential interpretations of flow dynamic  have to be explored with more detailed analyses when longer  time series are available. Furthermore, additional information on recharge processes will be extracted from analyses that include precipitation and discharge data.

**4 Synthesis**

In this work, we presented the objectives, measuring concept, and implementation of a soil moisture monitoring network at five representative sites across the globe to characterize karstic groundwater recharge and evapotranspiration processes. The monitoring network aims at improving the understanding (1) of the influence of soil and epikarst heterogeneities in karst regions on the flow and storage processes in the karst vadose zone, (2) of the impact of  land cover  on karstic groundwater recharge and evapotranspiration, and (3) of the qualitative climatic impacts on karstic groundwater recharge and evapotranspiration. In order to address these research objectives, >400

soil moisture probes were installed at five study sites located within different climatic regions. Each soil profiles consisted of 3 probes in 5 cm 10 cm and 80 cm depth (or at the soil bedrock interface, when the soils were shallower). In order to account for the effect of different land covers, the profiles at each of the five sites were equally split between a grassland plot and a forest plot, each with an extent of 20m by 20m. Yet limited to a few rainfall-soil moisture events sampled so far, our preliminary data already reveal different soil moisture responses at the different sites and land use types, such as sequential (top to bottom) reactions of soil moisture to rainfall events that can favour evapotranspiration, or non-sequential reactions indicating subsurface heterogeneity and preferential pathways that can result in enhanced groundwater recharge.

The wide range of local heterogeneity, land cover types and climate regions will allow new and detailed insights into the dynamics of groundwater recharge and evapotranspiration at the shallow subsurface in karstic regions. With longer records, additional data analysis will allow investigating the sequences of reaction of all available soil profiles (Demand et al. 2019), the identification of characteristic soil moisture states (Martini et al. 2015) and the quantitative assessment of soil hydraulic properties and evapotranspiration and recharge rates by physically-based soil models (Sprenger et al. 2015) will be possible. The measuring network will provide comprehensive data to develop and test new conceptual models of the functioning of the soil and epikarst conditional to climate and land use (Enemark et al. 2019) that will help to improve the realism of water resources models for karst regions (Mudarra et al. 2019), the quantification of land-use change effects on karstic recharge (Sarrazin et al. 2018), or the simulation of above-cave hydrology for improved speleothem paleoclimate reconstruction (Hartmann and Baker 2017, Cuthbert et al. 2014).

It is planned to maintain the monitoring network for terms longer than the funding time of this project (3 years) depending on future funding and local support. When several hydrological years are recorded, we will make our data publically available through a data publication in order to contribute to international efforts like the Critical Zone Observatories (Anderson et al. 2016),  the International Soil Moisture Network (Dorigo et al. 2011), or initiatives to improve global earth systems models (Fan et al. 2019).

**5   Author contributions**

The research and monitoring program was designed and implemented by RB with advice from AH, MM and MR. The paper was conceived by RB guided by AH and MR. BA, AB, DK, GL, AL, KL, MM, IY.P, FPA, RR and AV provided their local expertise before after and during the installation of the soil moisture network, as well as during the conception of the description of the study sites.

**6   Acknowledgments**

Romane Berthelin and Andreas Hartmann were supported by the Emmy Noether – Programme of the German Research Foundation (DFG; grant number HA 8113/1–1). Matías Mudarra and Bartolomé Andreo were supported by the Group RNM-308 funded by the Autonomous Government of Andalusia (Spain)  and Project CGL2015-65858-R founded by the General Office of Scientific and Technical Research (DGICYT) of Spanish Government. Rafael Rosolem is supported by the "A MUlti-scale Soil moisture Evapotranspiration Dynamics study" (AMUSED) [grant number NE/M003086/1] and the "Brazilian Experimental datasets for MUlti-Scale interactions in the critical zone under Extreme Drought" (BEMUSED) [grant number NE/R004897/1], both projects funded by Natural Environment Research Council (NERC). The article processing charge was funded by the German Research Foundation (DFG) and the University of Freiburg in the funding program Open Access Publishing. We thank Martin Maier for his advice concerning the soil description.